# The Cardiofaciocutaneous Syndrome: From Genetics to Prognostic–Therapeutic Implications

**DOI:** 10.3390/genes14122111

**Published:** 2023-11-22

**Authors:** Giovanna Scorrano, Emanuele David, Elisa Calì, Roberto Chimenz, Saverio La Bella, Armando Di Ludovico, Gabriella Di Rosa, Eloisa Gitto, Kshitij Mankad, Rosaria Nardello, Giuseppe Donato Mangano, Chiara Leoni, Giorgia Ceravolo

**Affiliations:** 1Department of Pediatrics, “G. D’Annunzio” University of Chieti-Pescara, 66100 Chieti, Italy; giovanna.scorrano@gmail.com (G.S.); armandodl@outlook.com (A.D.L.); 2Department of Translational and Precision Medicine, “Sapienza” University of Rome, 00161 Rome, Italy; emanueledavid@uniroma1.it; 3UCL Queen Square Institute of Neurology, University College London, London WC1N 3BG, UK; e.cali@ucl.ac.uk (E.C.); giorgiaceravolo@gmail.com (G.C.); 4Pediatric Nephrology and Dialysis Unit, University Hospital “G. Martino”, 98124 Messina, Italy; roberto.chimenz@polime.it; 5Child Neuropsychiatry Unit, Department of Human Pathology in Adult and Developmental Age “Gaetano Barresi”, University of Messina, 98124 Messina, Italy; gabriella.dirosa@unime.it; 6Neonatal and Pediatric Intensive Care Unit, Department of Human Pathology of the Adult and Developmental Age “Gaetano Barresi”, University of Messina, 98122 Messina, Italy; eloisa.gitto@unime.it; 7Department of Radiology, Great Ormond Street Hospital for Children, London WC1N 3JH, UK; drmankad@googlemail.com; 8Department of Health Promotion, Mother and Child Care, Internal Medicine and Medical Specialities, “G. D’Alessandro” University of Palermo, 90127 Palermo, Italy; rosaria.nardello@unipa.it (R.N.); giuseppedonato.mangano@unipa.it (G.D.M.); 9Center for Rare Diseases and Birth Defects, Department of Woman and Child Health and Public Health, Fondazione Policlinico Universitario A. Gemelli IRCCS, 00168 Rome, Italy

**Keywords:** Cardiofaciocutaneous syndrome, CFC, neurodevelopment, hypertrophic cardiomyopathy, RASopathies, MEK1 mutation, MEK2 mutation, BRAF mutation, KRAS mutation

## Abstract

Cardiofaciocutaneous (CFC) syndrome is one of the rarest RASopathies characterized by multiple congenital ectodermal, cardiac and craniofacial abnormalities with a mild to severe ocular, gastrointestinal and neurological involvement. It is an autosomal dominant syndrome, with complete penetrance, caused by heterozygous pathogenic variants in the genes *BRAF*, *MAP2K1/MEK1*, *MAP2K2/MEK2*, *KRAS* or, rarely, *YWHAZ*, all part of the RAS-MAPK pathway. This pathway is a signal transduction cascade that plays a crucial role in normal cellular processes such as cell growth, proliferation, differentiation, survival, metabolism and migration. CFC syndrome overlaps with Noonan syndrome, Costello syndrome, neurofibromatosis type 1 and Legius syndrome, therefore making the diagnosis challenging. Neurological involvement in CFC is more severe than in other RASopathies. Phenotypic variability in CFC patients is related to the specific gene affected, without a recognized genotype–phenotype correlation for distinct pathogenic variants. Currently, there is no specific treatment for CFC syndrome. Encouraging zebrafish model system studies suggested that, in the future, MEK inhibitors could be a suitable treatment of progressive phenotypes of CFC in children. A multidisciplinary care is necessary for appropriate medical management.

## 1. Introduction

CFC syndrome (OMIM 115150) is a multisystemic disorder that affects 1 in 800,000 newborns. CFC belongs to a group of syndromes collectively known as “RASopathies”, caused by germline mutations in one of the genes encoding components of the RAS-MAPK signal transduction pathway [1,2]. The RAS-MAPK pathway is a signal transduction cascade that plays a crucial role in normal cellular processes such as cell growth, proliferation, differentiation, survival, metabolism and migration. Somatic gain-of-function mutations in RAS genes are a well-known cause of cancer and there has been an increasing number of studies involving the RAS pathway in oncogenesis [3]. Though each syndrome has its specific phenotype, many of the RASopathies have overlapping clinical features, therefore making the diagnosis challenging. These can include facial dysmorphisms, intellectual disability (ID), congenital heart disease, short stature, muscle skeletal anomalies and lymphatic dysfunction. CFC syndrome may clinically overlap with neurofibromatosis type 1 (NF1), Noonan syndrome (NS), Noonan syndrome with multiple lentigines (NS-ML), Costello syndrome (CS), Legius syndrome (LS), central conducting lymphatic anomalies syndrome (CCLA), SYNGAP1 syndrome, and capillary malformation arteriovenous malformation syndrome (CM-AVM), though compared with other RASopathies, it exhibits more severe neurologic complications and phenotype [4]. CFC patients frequently present with a characteristic facial appearance, congenital heart disease, severe developmental delay and psychomotor delay, ectodermal, ocular and gastrointestinal abnormalities, and neurological involvement [1]. Characteristic facial features include high forehead, bitemporal constriction, supraorbital hypoplasia, downslanting palpebral fissures, depressed nasal bridge and posteriorly rotated ears with thick helices [5,6]. Compared to other RASopathies, neurological involvement is more severe and commonly includes refractory epilepsy, neurocognitive impairment, motor and speech deficits, hypotonia and behavioral challenges; ventriculomegaly, cortical atrophy and hydrocephalus are the most frequent imaging findings in CFC patients [6,7]. Cardiac involvement is common in CFC and manifests as atrial and ventricular septal defects, hypertrophic cardiomyopathy and pulmonary valve stenosis. These patients also have curly and friable hair, and sparse eyebrows and eyelashes; skin involvement ranges from dry skin to hyperkeratosis. CFC is inherited in an autosomal-dominant manner, and most pathogenic variants involve the genes *BRAF* (75%), *MAP2K1* (25%), *MAP2K2* (25%), *KRAS* (2%) or, more rarely, *YWHAZ* [8]. The real incidence is still unknown and a phenotypic variability related to the specific gene affected has been identified, without a genotype–phenotype correlation for specific pathogenic variants (Table 1) [5]. The diagnosis of CFC syndrome is suspected based on the distinctive clinical features and it is confirmed by molecular genetic testing [5,9]. The aim of the study is to give updated evidence on the pathogenesis, clinical and radiological features, diagnosis, and available treatment for CFC, supplying practice guidelines for the management of pediatric patients with CFC.

## 2. Methodology

In this narrative review, we searched articles on PubMed up to September 2023. The search formula was as follows: (((CFC) OR (Cardiofaciocutaneous Syndrome)) AND ((gene pathogenic variants) OR (mutations) OR (related neurological disorders) OR (neuroradiological findings) OR (related skin anomalies) OR (related gastrointestinal disorders) OR (muscle-skeletal findings) OR (behavior and cognitive disorders) OR (cardiac disorders) OR (craniofacial findings) OR (endocrinologic and growth disorders))).

We selected studies, published in English, that included CFC pathogenesis, clinical and radiological features, diagnosis, and the latest evidence on new treatments.

We revised the literature, exploring the evolving spectrum related to pathogenic variants affecting patients with CFC, described to date. Of note, we firstly reported systematically the phenotypic variability related to the specific gene affected in the syndrome, with relevant prognostic–therapeutic implications. Additionally, we first mentioned in the review the new gene causally related to CFC, *YWHAZ*, describing how de novo missense pathogenic variants in this gene lead to RAS/MAPK pathway disruption and clinical manifestations. The family involved in the study provided informed consent for being part of this study.

## 3. Pathogenesis

Currently, *BRAF*, *MAP2K1*, *MAP2K2* and, rarely, *KRAS* have been associated with CFC encoding components of the RAS-MAPK pathway. The signal cascade of RAS-MAPK is highly conserved and plays a crucial role in normal cellular processes as cell growth, proliferation, differentiation, survival, metabolism and migration [5,10]. The pathway is stimulated by growth factors that lead RAS-mediated RAF activation. This kinase phosphorylates MEK1 and/or MEK2 with consequent ERK1/2 activation, downstream the cascade [11]. ERK1 and ERK2 effectors act both in the nucleus and in the cytoplasm, mediating the polymorphic cellular response to growth factors (Figure 1).

Pathogenic variants are mainly missense with a gain-of-function mechanism on proteins of the pathway, leading to ERK1-2 hyperactivation.

The RAS-MAPK pathway is a signal transduction cascade that plays a crucial role in normal cellular processes such as cell growth, proliferation, differentiation, survival, metabolism and migration. The pathway is stimulated by growth factors that lead to RAS-mediated RAF activation. This kinase phosphorylates MEK1 and/or MEK2 with consequent ERK1/2 activation, downstream the cascade. ERK1 and ERK2 effectors act both in nucleus and in cytoplasm, mediating the polymorphic cellular response to growth factors.

## 4. Neurological Findings

Neurological involvement in CFC syndrome is present in nearly all individuals and ranges from mild to severe. Developmental delay, ID, epilepsy and hypotonia are common findings, although a few individuals may have normal cognitive function (IQ within normal range).

In children with CFC, seizures frequently occurred, usually with onset in infancy or early childhood and they included tonic–clonic and/or complex focal seizures, absence seizures and epileptic infantile spasms [7,12,13,14,15,16]; seizures could occur later in infancy, requiring polytherapy and refractoriness to antiseizure medications (ASMs) [5]. However, in children, seizure may occur also due to multiple different genetic conditions [17,18,19,20].

Hypotonia is also described and presented typically with a neonatal onset with mild to severe asthenia, motor delay and poor muscle mass [5]. Furthermore, behavior abnormalities could characterize a neuropsychiatric disorder in CFC patients, including irritability, obsessive compulsive disorder, anxiety and autistic traits [5,21].

A recent clinical study by Pierpont et al. [6] reported a higher rate of CFC patients affected by epilepsy than in the previously published literature: seizures of various type occurred in 57% of individuals with *BRAF* pathogenic variants, 61% of patients with *MAP2K1* pathogenic variants and 30% of subjects with *MAP2K2* pathogenic variants. *MAP2K2* mutations were associated with a lower risk of seizure occurrence and less severe seizure types [6]. On the other hand, patients with pathogenic variants in *BRAF* and *MAP2K1* presented with moderate to severe polymorphic seizures, which were frequently drug-resistant. Furthermore, a relevant study by Battaglia et al. [16] shed a light on genotype–phenotype correlation concerning the epileptic phenotype. There seems to be a strong genotype–phenotype correlation regarding the occurrence and severity of seizures, since variants in the protein kinase domain of *BRAF* (exons 11–16) and the p.Y130C/H/N variant of *MAP2K1* (exon 3) correlate with a severe epileptic phenotype [6]. Nevertheless, pathogenic variants in *KRAS* variants were not associated with epilepsy. Notably, most children affected with the above genotypes and severe epilepsy were also presenting neurodevelopmental impairment. ID was reported in most individuals (82 -100%), usually carrying *BRAF* (89%) or *MAP2K1* (84%) pathogenic variants, while it was described only in 25% of patients with *MAP2K2* pathogenic variants [6]. The degree of ID ranged from mild (*MEK*) to moderate (*BRAF*) [22]. Neurodevelopmental delay may be mild and difficult to notice in patients with mild or moderate involvement, while speech and motor delays are usually obvious in severely affected children [23]. Nevertheless, those are non-specific findings, often found also in children with other kinds of neurologic impairment [19,24,25,26]. Motor delay and hypotonia are the most frequently observed neurological features in CFC syndrome [27]. Pierpont et al., reported a distinctive genotype–phenotype correlation: children carrying pathogenic variants in *MAP2K1* showed more need for support and lack of independent ambulation (71%) compared to patients with *BRAF* (29%) or *MAP2K2* (13%) mutations [6]. Language acquisition is usually delayed and speech abilities range from full sentences to non-verbal communications (9–31%) [9]. Interestingly, mutations promoting dysregulation of the RAS-MAPK cascade have been associated with an increased psychopathological risk. Frequently, patients with CFC presented underdiagnosed autistic-like behaviors [21].

A magnetic resonance imaging (MRI) evaluation could be useful during the diagnostic work-up. Brain magnetic resonance imaging studies may prove useful in patients with uncertain diagnosis and for prognostic information. Ventriculomegaly (43.9%) is the most common finding and sometimes requires a ventriculo-peritoneal shunt [28,29].

Other findings comprise prominent Virchow–Robin spaces (10.6%), hydrocephalus (24.2%) and myelination abnormalities. Some patients have structural brain anomalies, including Type I Chiari malformation (4.5%), arachnoid cyst (1.5%) and subependymal grey matter heterotopia (7.5%) [12].

Other abnormalities can include abnormal EEG, dilated perivascular spaces, dilated perivascular spaces, corticospinal tract findings, agenesis of the corpus callosum (6%), frontal lobe hypoplasia (1.5%), pachygyria (7.5%) and Chiari malformation (4.5%) [5] (Figure 2).

### 4.1. Cardiac Findings

Congenital heart defects (CHDs) occur in most people with CFC syndrome (~75%) [9,30,31,32]. The cardiac defects most frequently associated with this condition include pulmonary valvular stenosis (PVS) (~45%), Hypertrophic cardiomyopathy (HCM) (~40%), atrial septal defect (18–28%), ventricular septal defect (11–22%) and other anomalies such as mitral valve dysplasia, arrhythmias, tricuspid valve dysplasia, and bicuspid aortic valve [9]. Patients with *MAP2K1* variants have a lower frequency of cardiac disease than those with *MAP2K2* (64%) and *BRAF* variants (72%) [8]. Literature data show that neurological and cardiac phenotypes do not segregate in a similar pattern, so different variants in CFC genes have different consequences across tissues [6].

### 4.2. Ectodermal Findings

Ectodermal anomalies are cardinal features of CFC syndrome; virtually all patients with CFC syndrome develop some kind of ectodermal involvement.

The most common manifestations are sparse, curly, fine, brittle slow-growing hair; sparse to absent eyebrows with ulerythema ophryogenes; sparse to absent eyelashes; dystrophic rapid-growing nails; skin abnormalities such as keratosis pilaris, hyperkeratosis, ichthyosis, eczemas, xerosis, hemangiomas and numerous pigmented naevi (more common in patients with *BRAF* mutations) [33,34,35,36]. Some skin anomalies evolve with age: xerosis and follicular hyperkeratosis can improve [37], palmoplantar hyperkeratosis tends to worsen, especially in pressure areas, lymphedema may become more severe, and pigmented naevi [38,39] tend to grow in number [40]. It is also common for CFC patients to be affected by severe skin infections [5].

### 4.3. Craniofacial Findings

Children with CFC syndrome have characteristic craniofacial features. Those include macrocephaly, high forehead, bitemporal constriction, hypoplasia of the supraorbital ridges, downslanting palpebral fissures (more frequent in patients with *MEK* mutations than *BRAF*), ocular hypertelorism (more commonly associated with mutations in *BRAF*), ptosis, epicanthal folds, telecanthus, short nose with depressed bridge and anteverted nares, low-set and posteriorly angulated ears with prominent helices, high-arched palate and relative micrognathia [9].

### 4.4. Gastrointestinal and Growth Findings

Feeding problems are very frequently reported, almost in all CFC patients, and there does not seem to be any genotype–phenotype correlation [41,42]. These problems are usually reported in the neonatal period and manifest as poor suck, swallow dysfunction, vomiting, gastroesophageal reflux, aspiration, and oral aversion. Feeding issues are usually followed by inadequate oral caloric intake, failure to thrive, and often require nasogastric tube feeding (40%) or gastrotomy tube placement (50%) [9]. In these children, growth is usually delayed with both weight and length below normal range [5]. Oral aversion can persist during childhood but usually GI symptoms improve with age, other than in patients with epilepsy onset in childhood. Their GI disorders become worse with age. Constipation is often seen in children with CFC, other findings include intestinal malrotation, hernia and also splenomegaly, hepatomegaly and steatosis. Many common symptoms are a consequence of dysmotility including gastroesophageal reflux, vomiting and constipation (Figure 3) [22].

It is important to notice that, though the short stature could be related to a growth delay caused by poor feeding, it could also be linked to a growth hormone deficiency, since the RAS-MAPK pathway plays an important role in the insulin-like growth factor I (IGF-I) mediations and growth hormone (GH) secretion.

### 4.5. Additional Features

During perinatal period polyhydramnios (77%) and prematurity (up to 50%) have been reported as common findings [9,22].

The vast majority of individuals present with musculoskeletal abnormalities including hypotonia, pectus excavatum and carinatum, scoliosis, kyphosis, short neck and pterygium coli. The orthopedic features are more common in patients with *MAP2K1* variant than in patients with other variants according to Leoni et al. [43]. Interestingly, patients with CFC present a reduced bone mineral density (BMD), probably associated with reduced physical activity and inflammatory cytokines. Despite vitamin D supplementation and almost normal bone metabolism biomarkers, CFC patients showed significantly decreased absolute values of DXA-assessed subtotal and lumbar BMD (*p* ≤ 0.05), compared to controls, with BMD z-scores and t-scores below the reference range in CFC, and normal in healthy controls [44].

Endocrine complications have been evaluated in patients with CFC. Specifically, a high prevalence of thyroid autoimmunity, with an increased risk to develop autoimmune disorders, and short stature have been described and presumably related to the dysregulation of the RAS-MAPK pathway, the reduced physical activity, the presence of inflammatory cytokines and the impaired IGF1 activity [45].

Ocular manifestations including strabismus, nystagmus, ocular hypertelorism, astigmatism, optic nerve hypoplasia, myopia, hyperopia, cortical blindness, and cataracts are present in most individuals with CFC syndrome and may result in poor visual acuity. Optic nerve hypoplasia is more commonly reported in patients with *BRAF* mutations [22,46].

Notably, functional limitations, pain and disability have been reported in patients with CFC. Using the Pediatric Outcomes Data Collection Instrument (PODCI) and Pediatric Evaluation of Disability Index (PEDI), an interesting study [27] documented that CFC patients presented activity limitations in the PODCI domains of upper extremity function, transfers and mobility, sport, and physical function. Concurrently, a relevant disability in the PEDI domains of daily activity, mobility, socialization and cognition was noted. Pain is highly prevalent in patients with RASopathies and CFC, as well. Specifically, musculoskeletal and abdominal pain was more frequently reported and often interfered with daily activities. Furthermore, pain negatively impacted QoL and sleeping patterns [47].

Interestingly, neoplasms, commonly observed in other RASopathies, have not been reported in CFC syndrome, even though genetic mutations causing CFC syndrome are well known to play a crucial role in development and oncogenesis. *BRAF* mutations have been associated with 7% of cancers [48,49], and there are few published reports of neoplasia in CFC [23,50], with only one malignancy [51,52].

Urogenital anomalies may occur in ~33% of individuals with CFC: cryptorchidism is the most common feature (66% of affected males); also, renal cysts, nephrolithiasis and bladder abnormalities may be present.

## 5. Diagnosis

The diagnosis of CFC syndrome is often challenging, and it should be based on a specific clinical manifestation and confirmed through molecular genetic testing.

The most common clinical features are short stature, distinctive craniofacial features, developmental delay, neuromotor delay, ID, ectodermal abnormalities, congenital heart disease, musculoskeletal features, feeding and gastrointestinal problems, and cryptorchidism. Features of CFC syndrome overlap considerably with some other RASopathies, including Costello syndrome, and Noonan syndrome. Some features may be present since birth while others may appear later in life; also, these clinical conditions can change over time either by becoming better or becoming more severe. Phenotypic variability in CFC is also associated with different variants in CFC genes. These phenotypic changes, overlaps and variability create diagnostic difficulties [6].

CFC syndrome is an autosomal-dominant syndrome, with complete penetrance [5,23].

An affected person has a 50% chance of passing the altered gene to each child. The mutation genes involved in CFC syndrome are *BRAF* (7q34) in up to 75% of the cases, *MAP2K1* (15q22.1-q22.33) and *MAP2K2* (19p13.3), both around 25% of the cases, and *KRAS* (12p12.1) present in less than 2% of the cases. Most individuals with CFC syndrome reported to date have a de novo BRAF, MAP2K1, MAP2K2, or KRAS pathogenic variant. Even though CFC is an autosomal dominant disorder, sporadic cases are the majority. To date, only seven families were reported as characterized by a vertical transmission of MEK2, KRAS, and BRAF pathogenic variants, respectively. This suggests that CFC pathogenic variants within the Ras/MAPK pathway are compatible with human reproduction. Additionally, the reproductive success may be affected by CFC phenotypic features, such as neurodevelopmental delay, instead of the disruption of the Ras/MAPK pathway. An intrafamilial variable expressivity has been identified, as well [53].

Recently, *YWHAZ*, a 14-3-3 family member, has been identified as a new gene involved in CFC. It was documented that the S230W *YWHAZ* variant enhanced Raf-stimulated Erk phosphorylation to a higher level than wild type, with a gain-of-function mechanism in the RAS-ERK pathway. This result supported that the variant was the underlying cause of the CFC phenotype [8]. Mutations in these genes cause a dysregulation of the RAS/MAPK signaling pathway. This pathway is critically involved in cell differentiation, proliferation, migration, and apoptosis, and is one of the most studied signaling cascades.

The genetic diagnosis of CFC syndrome is established by the identification of a heterozygous pathogenic variant in *BRAF*, *MAP2K1*, *MAP2K2*, or *KRAS* genes.

### Current Consensus Guidelines Strategy Include the Following

Multigene panel for common RASopathy genes that includes *BRAF*, *MAP2K1*, *MAP2K2*, *KRAS* and *YWHAZ* usually detects up to 90% of individuals with CFC and it is the preferred initial test.Individual single-gene testing is recommended if panel testing is not available, beginning with *BRAF*, *MAP2K1*, and *MAP2K2*, and *KRAS*.If these molecular genetic tests are negative, a more comprehensive genetic sequencing including exome and genome sequencing should be performed [5].

In the event that a pathogenic variant is identified, prenatal testing should be recommended to determine if the mutations is inherited [5,54].

## 6. Treatment

Currently, there is no specific treatment in CFC syndrome. More recently, genetic pathway inhibitors, including MEK inhibitors, led to size reduction of inoperable plexiform neurofibromas (PNs) in NF1, with fewer side effects [55]. Moreover, in several mouse or other model systems, MEK inhibitors were used as a prenatal preventative therapy or postnatal treatment of non-NF1 RASopathies. In this context, in the in vivo zebrafish model system, it was observed that CFC mutations in *BRAF* and MEK impaired convergence–extension cell movements during gastrulation and MEK inhibitors prevented the cell migration defects caused by this pathogenetic variants in CFC, treating the embryos within a specific developmental time-window [11]. Additionally, an interesting functional study using human dermal lymphatic endothelial cells (HDLECs) and zebrafish larvae, to model *KRAS* pathogenic variants, documented an increased ERK phosphorylation with biochemical and morphological changes, significantly rescued by the subsequent use of MEK inhibitors [56]. Interestingly, two studies revealed how MEK inhibitors represented a promising treatment in HCM related to CFC syndrome, especially if administered before the onset of irreversible cardiac remodeling [57,58].

Moreover, a trial focusing on use of a MEK1/2 inhibitor, Trametinib, in NF1 or GNA11/Q alterations, did not demonstrate meaningful clinical activity. However, it appeared effective in one patient with multiple cancers, warranting future studies [59]. Furthermore, it was observed that Braf+/+ and BrafQ241R/+ mice treated with C-type natriuretic peptide (CNP), a stimulator of endochondral bone growth and a potent inhibitor of the FGFR3-RAF1-MEK/ERK signaling, presented increased body and tail lengths, suggesting that CNP could be a potential therapeutic target in CFC syndrome [60]. Additionally, BrafQ241R/+ knock-in mice showing cardiovascular and lymphatic defects ameliorated after combination treatment with a MEK inhibitor, PD0325901, and a histone 3 demethylase inhibitor, GSK-J4, suggesting that epigenetic modulation as well as the inhibition of the ERK pathway could be a potential therapeutic strategy [61].

These results suggest that small molecule inhibitors could be potentially used to treat the progressive phenotypes of CFC in children. Hence, future pre-clinical models of CFC should be developed, to better define protein interactions of the RAS/MAPK pathway, the pathophysiology of the disorder, and endpoints for measuring treatment efficacy [62].

Multidisciplinary care is crucial in CFC children; despite this, there is no specific treatment. Specifically, a comprehensive management should be performed, especially for individuals at risk. If patients are at risk for pulmonary stenosis, HCM and other cardiac defects, they should be submitted to a cardiac follow-up with blood pressure measurement at each visit and echocardiogram every 2–3 y up to 20 years of age, if no cardiac disease is found initially. Concurrently, frequent dermatology visits for the management of skin abnormalities such as keratosis pilaris, hyperkeratosis, ichthyosis, eczemas, xerosis, hemangiomas and annual evaluation of pigmented naevi should be performed. Furthermore, individuals at risk for infantile spasms, seizures, and brain anomalies should be evaluated from neurologists in a continuous follow-up. Functional behavior assessment, special education services, and early childhood intervention programs should be considered in patients with cognitive and behavior disorders, as well. If gastrointestinal difficulties are present, a regular follow-up to monitor growth and nutrition should be performed, with the measurement of growth parameters and nutritional status at each visit. Moreover, ophthalmologic defects should be controlled every six to 12 months, whereas audiologic evaluation every two to three years. Concurrently, patients with musculoskeletal anomalies, genitourinary malformations, hematologic disorders and dental lesions should be systematically followed by specialists to prevent complications, improving the quality of life of these complex patients [5,9].

## 7. Conclusions

In recent years, studies based on NGS and omics-related sciences revealed an expanding molecular complexity underlying genetic disorders such as RASopathies [63,64,65,66,67]. Many novel molecular factors and genes have been identified with consequent benefits in terms of refining clinical phenotypes, valuable prognostic information, detailed imaging studies and targeted therapies for the children affected with these conditions [68,69,70,71,72]. The CFC phenotype is very heterogeneous, as it depends on the gene involved, as well as the type and localization of the pathogenic variant, and it can change and evolve across time. Like other RASopathies, CFC syndrome presents distinctive clinical features and involves numerous organ systems. Even though, there is no specific treatment in CFC syndrome, encouraging zebrafish model system studies suggested that MEK inhibitors could be a suitable targeted treatment of progressive phenotypes of CFC in future. Currently, a multidisciplinary care is necessary to manage the numerous issues that are present in this condition: neurological, cardiological, gastroenterological, endocrinological, dermatological, orthopedic, ophthalmological, and behavioral problems. Recent clinical studies highlighted the strong and significant correlations between different phenotypes and gene variants. Investigations to better understand the disease pathway in this pediatric population is needed and it will hopefully enable and guide more effective therapeutic avenues in the future.

## Figures and Tables

**Figure 1 genes-14-02111-f001:**
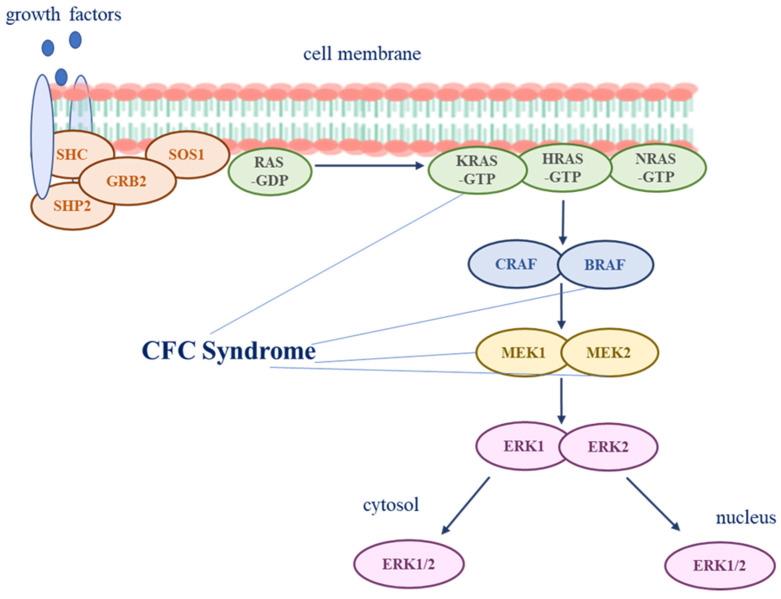
The RAS-MAPK pathway.

**Figure 2 genes-14-02111-f002:**
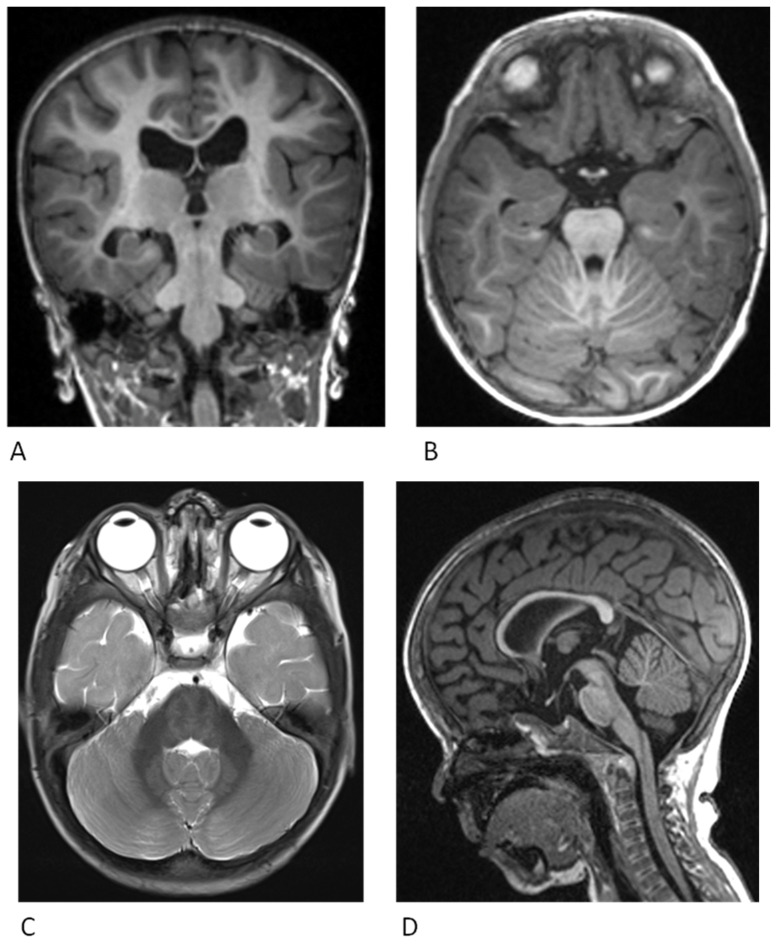
Neuroradiological findings in patient with CFC. Magnetic resonance imaging (MRI) scan of the brain of a 2-year-old patient with a *MAP2K1* variant in CFC Syndrome. (**A**) Sagittal T1: Reduced cerebral volume anteriorly with associated thinning of the corpus callosum. (**B**) Coronal T1: Bilateral hippocampal malrotation. (**C**) Axial T1: Delayed myelination within the temporal lobes. (**D**) Axial T2: Abnormal signal within the dentate nuclei.

**Figure 3 genes-14-02111-f003:**
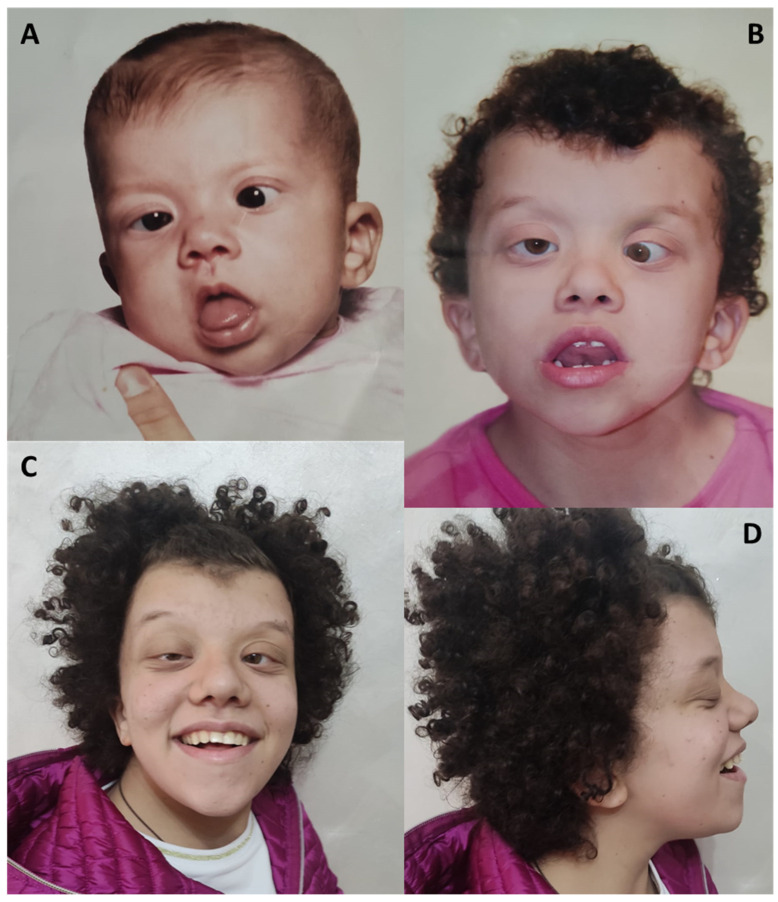
Distinctive craniofacial features in a CFC patient through time. At 12 months (**A**), 5 years (**B**) and 16 years of age (**C**,**D**), respectively.

**Table 1 genes-14-02111-t001:** Genetic findings in patients with CFC.

Gene	OMIM Number	Prevalence	Inheritance	CFC Phenotypic Features
*BRAF*	*164757	75%	De novo missense heterozygous variants Autosomal-dominant missense variant inherited by mother (1 case)	Moderate to severe polymorphic seizures (+exons 11–16) 57% Moderate ID 89% Motor delay and hypotonia (unable to walk, needing support) 29% Cardiac disease 72% Greater risk of skin abnormalities Ocular hypertelorism, optic nerve hypoplasia Tumors (+melanoma, thyroid, colorectal, and ovarian cancers, benign nevi, premalignant colon polyps 8% Pulmonary stenosis 50%
*MAP2K1*	*176872	25%	De novo missense heterozygous variants	Moderate to severe polymorphic seizures (+p.Y130C/H/N variant, exon 3) 61% Mild ID 84% Motor delay and hypotonia (unable to walk, needing support) 71% Cardiac disease (lower frequency, NA) Macrocephaly, high forehead, bitemporal constriction, hypoplasia of the supraorbital ridges, downslanting palpebral fissures Musculoskeletal abnormalities
*MAP2K2*	*601263	25%	De novo missense heterozygous variants Autosomal dominant missense variants inherited by mother (4 cases)	Lower risk of seizure occurrence and less severe seizure types 30% Mild ID 25% Motor delay and hypotonia (unable to walk, needing support) 13% Cardiac disease 64% Macrocephaly, high forehead, bitemporal constriction, hypoplasia of the supraorbital ridges, downslanting palpebral fissures
*KRAS*	*190070	2%	De novo missense heterozygous variants Autosomal dominant missense variants inherited by mother (2 cases)	No epilepsy Neurodevelopmental delay Coarse face Cardiac defects
*YWHAZ*	*601288	Rarely, NA	De novo missense heterozygous variants	Developmental delay, behavioural disorders ID Short stature Motor and speech delay Triangular facies, ptosis Seizures Feeding problems

## Data Availability

Data are contained within the article.

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
