# Peer review of "The Cardiofaciocutaneous Syndrome: From Genetics to Prognostic–Therapeutic Implications"

_genes, 2023, doi:10.3390/genes14122111_

Round 1

Reviewer 1 Report

Comments and Suggestions for Authors

The authors offer a comprehensive review of cardiofaciocutaneous syndrome (CFC), including the affected genetic pathway, genotype-phenotype correlations, key clinical features, and potential future targeted therapies. Overall the manuscript is well-written and researched with appropriate citations throughout.

Some specific comments:

Lines 61-63 - if example patient photographs are available, they would add significantly to the manuscript.

Line 73 - "no phenotype-genotype correlation has been identified..." - yet you cite various phenotype-genotype correlations throughout the manuscript (e.g. lines 118-121; 132-135; 165-166; etc.)

Line 86 - "small deletions with a gain of function mechanism" - are these deletions in-frame? Otherwise they would be expected to result in a loss of function (due to frameshift and premature termination).

Line 96 - although you present some radiological findings, much of this section is not imaging related; consider making the heading simply "Neurological findings".

Line 140 - "A correct diagnosis requires MRI evaluation"- is this necessary to diagnosis CFC, or does genetic testing and other clinical features suffice? Perhaps you mean that brain imaging is recommended to characterize fully any CNS abnormalities.

Line 252 - "confirmed trough molecular testing" - I am not certain what is meant by "trough" molecular testing.

Line 287 (Treatment) - although not a specific treatment for CFC, it may be useful to include clinical care guidelines for individuals with a diagnosis of CFC (e.g. brain imaging, echocardiogram, developmental assessment, etc.)

Comments on the Quality of English Language

Only a few awkward phrasings; overall pretty good

Author Response

Reviewer 1: The authors offer a comprehensive review of cardiofaciocutaneous syndrome (CFC), including the affected genetic pathway, genotype-phenotype correlations, key clinical features, and potential future targeted therapies. Overall, the manuscript is well-written and researched with appropriate citations throughout.

1)Lines 61-63 - if example patient photographs are available, they would add significantly to the manuscript.

Response: Thank you very much for your suggestion. We inserted representative photos of CFC patients.

2) Line 73 - "no phenotype-genotype correlation has been identified..." - yet you cite various phenotype-genotype correlations throughout the manuscript (e.g. lines 118-121; 132-135; 165-166; etc.).

Response: Thank you for your comment. We reformulated the statement as follows: “The real incidence is still unknown and a phenotypic variability related to the specific gene affected has been identified, without a genotype-phenotype correlation for specific pathogenic variants”. Even thought, we haven’t found a recognised phenotype-genotype correlation for specific pathogenic variants in BRAF, KRAS, MAP2K1, YWHAZ or MAP2K2, however, a phenotypic variability related to the specific gene affected has been identified, as we described both in the manuscript and in the Table 1 (e.g. “MAP2K2 mutations were associated with a lower risk of seizure occurrence and less severe seizure types, whereas patients with pathogenic variants in BRAF and MAP2K1 presented with moderate to severe polymorphic seizures, frequently drug resistant.”)

3) Line 86 - "small deletions with a gain of function mechanism" - are these deletions in-frame? Otherwise, they would be expected to result in a loss of function (due to frameshift and premature termination).

Response: We are grateful for your comment, we corrected the sentence. In classical CFC syndrome, deletions were not detected. However, MEK2 microdeletions with a gene haploinsufficiency were reported in three studies, related to a phenotype including a characteristic craniofacial appearance and pattern of malformation and organ dysfunction. Functional studies revealed a dis-regulation of Ras/MAPK pathway and authors speculated a new model of a non-classical RASopathy related to MEK2 haploinsufficiency (doi 10.1111/cge.12116, doi 10.1002/ajmg.a.33986, doi 10.1002/ajmg.a.33738).

4) Line 96 - although you present some radiological findings, much of this section is not imaging related; consider making the heading simply "Neurological findings".

Response: Thank you for your comment. We changed the heading, as you suggested.

5) Line 140 - "A correct diagnosis requires MRI evaluation"- is this necessary to diagnosis CFC, or does genetic testing and other clinical features suffice? Perhaps you mean that brain imaging is recommended to characterize fully any CNS abnormalities.

Response: Thanks, we reformulated the sentence. Indeed, the CFC diagnosis requires specific clinical manifestation, confirmed through molecular genetic testing.

6) Line 252 - "confirmed trough molecular testing" - I am not certain what is meant by "trough" molecular testing.

Response: Thank you for your comment. It was a typing error. We corrected this as follows: “through”.

7) Line 287 (Treatment) - although not a specific treatment for CFC, it may be useful to include clinical care guidelines for individuals with a diagnosis of CFC (e.g. brain imaging, echocardiogram, developmental assessment, etc.)

Response: Thank you for your advice. We inserted in the “Treatment section” a brief description of the surveillance needed for patients with CFC.

Reviewer 2 Report

Comments and Suggestions for Authors

General comment

I am pediatrician and pediatric cardiologist. I am running a case-series project in which we use a MEK inhibitor in hypertrophic cardiomyopathy in Noonan syndrome. That’s why my expertise will mainly focus on this aspect of this article. 

This is a well-written article about Cardiofaciocutaneous (CFC) syndrome. I assume that the authors' objective was to summarize the current knowledge on the pathogenesis, diagnosis, and available treatment of this syndrome. However, my main concerns are about the aim of this article which should be stated by the authors in the abstract and manuscript of this article. If the main reason for this article was to summarize current knowledge, the authors should have performed it as a systematic review or at least a rapid review of each of the domains (e.g. for the diagnosis, treatment, etc.) with a brief methodology description. According to some recent narrative reviews on this topic:

https://www.ncbi.nlm.nih.gov/books/NBK1186/

https://www.ncbi.nlm.nih.gov/pmc/articles/PMC4179092/

I am not sure what this article adds to current knowledge. The authors should have stated what was the main reason for writing such an article and what it adds to status because it is not clear to me. 

Running a rapid review on this topic (with proper methodology: https://ebm.bmj.com/content/early/2023/04/19/bmjebm-2022-112079) would improve such work. 

Minor Comments:

·      Abstract

I would add information that CFC syndrome is one of the rarest Rasopathies or I would add incidence of CFC or other Rasopathies

In the sentence “Phenotypic variability in CFC patients is strongly associated with different variants in these genes” I am not sure to what genes it refers to, probably this sentence should be used earlier in the article. 

Please add what was the main reason for writing such an article and state what is new for the readers.

·      Neuroradiological findings

This is a well-described paragraph with a very large Figure 2. However, I am confused as to why is there such a large disproportion between the remaining paragraphs. Is that because the neurological changes are much more common in CFC syndrome? 

If the authors' aim was to summarize the clinical characteristics of CFC syndrome, I would suggest firstly balancing the volume of individual chapters and adding instead of figure 3 a real photo of patients CFC syndrome in different age (if it is possible for the authors) 

·      Current Consensus guidelines strategy include the following

Line 288 – “[4].” Should be without bold.

·      Treatment

This paragraph needs some improvements too. First of all, authors did not use up-to-date references according to NF1 and KRAS patients (some suggestions below). Secondly, I would focus more on studies in which the efficacy of MEK inhibitors was assessed in childrens or adults 

Wisinski KB, Flamand Y, Wilson MA, Luke JJ, Tawbi HA, Hong F, Mitchell EP, Zwiebel JA, Chen H, Gray RJ, Li S, McShane LM, Rubinstein LV, Patton D, Williams PM, Hamilton SR, Behrens RJ, Pennington KP, Conley BA, Arteaga CL, Harris LN, O'Dwyer PJ, Chen AP, Flaherty KT. Trametinib in Patients With NF1-, GNAQ-, or GNA11-Mutant Tumors: Results From the NCI-MATCH ECOG-ACRIN Trial (EAY131) Subprotocols S1 and S2. JCO Precis Oncol. 2023 Apr;7:e2200421.

Sheppard SE, March ME, Seiler C, Matsuoka LS, Kim SE, Kao C, Rubin AI, Battig MR, Khalek N, Schindewolf E, O'Connor N, Pinto E, Priestley JR, Sanders VR, Niazi R, Ganguly A, Hou C, Slater D, Frieden IJ, Huynh T, Shieh JT, Krantz ID, Guerrero JC, Surrey LF, Biko DM, Laje P, Castelo-Soccio L, Nakano TA, Snyder K, Smith CL, Li D, Dori Y, Hakonarson H. Lymphatic disorders caused by mosaic, activating KRAS variants respond to MEK inhibition. JCI Insight. 2023 May 8;8(9):e155888. doi: 10.1172/jci.insight.155888. PMID: 37154160; PMCID: PMC10243805.

Maeda Y, Tidyman WE, Ander BP, Pritchard CA, Rauen KA. Ras/MAPK dysregulation in development causes a skeletal myopathy in an activating BrafL597Vmouse model for cardio-facio-cutaneous syndrome. Dev Dyn. 2021 Aug;250(8):1074-1095. doi: 10.1002/dvdy.309. Epub 2021 Feb 13. PMID: 33522658.

Rauen KA, Alsaegh A, Ben-Shachar S, Berman Y, Blakeley J, Cordeiro I, Elgersma Y, Evans DG, Fisher MJ, Frayling IM, George J, Huson SM, Kerr B, Khire U, Korf B, Legius E, Messiaen L, van Minkelen R, Nampoothiri S, Ngeow J, Parada LF, Phadke S, Pillai A, Plotkin SR, Puri R, Raji A, Ramesh V, Ratner N, Shankar SP, Sharda S, Tambe A, Vikkula M, Widemann BC, Wolkenstein P, Upadhyaya M. First International Conference on RASopathies and Neurofibromatoses in Asia: Identification and advances of new therapeutics. Am J Med Genet A. 2019 Jun;179(6):1091-1097. doi: 10.1002/ajmg.a.61125. Epub 2019 Mar 25. PMID: 30908877; PMCID: PMC8279388.

Andelfinger G, Marquis C, Raboisson MJ, Théoret Y, Waldmüller S, Wiegand G, Gelb BD, Zenker M, Delrue MA, Hofbeck M. Hypertrophic Cardiomyopathy in Noonan Syndrome Treated by MEK-Inhibition. J Am Coll Cardiol. 2019 May 7;73(17):2237-2239. doi: 10.1016/j.jacc.2019.01.066. PMID: 31047013; PMCID: PMC6916648.

Mussa A, Carli D, Giorgio E, Villar AM, Cardaropoli S, Carbonara C, Campagnoli MF, Galletto P, Palumbo M, Olivieri S, Isella C, Andelfinger G, Tartaglia M, Botta G, Brusco A, Medico E, Ferrero GB. MEK Inhibition in a Newborn with RAF1-Associated Noonan Syndrome Ameliorates Hypertrophic Cardiomyopathy but Is Insufficient to Revert Pulmonary Vascular Disease. Genes (Basel). 2021 Dec 21;13(1):6. doi: 10.3390/genes13010006. PMID: 35052347; PMCID: PMC8774485.

Author Response

Reviewer 2. I am pediatrician and pediatric cardiologist. I am running a case-series project in which we use a MEK inhibitor in hypertrophic cardiomyopathy in Noonan syndrome. That’s why my expertise will mainly focus on this aspect of this article.

This is a well-written article about Cardiofaciocutaneous (CFC) syndrome. I assume that the authors' objective was to summarize the current knowledge on the pathogenesis, diagnosis, and available treatment of this syndrome. However, my main concerns are about the aim of this article which should be stated by the authors in the abstract and manuscript of this article. If the main reason for this article was to summarize current knowledge, the authors should have performed it as a systematic review or at least a rapid review of each of the domains (e.g. for the diagnosis, treatment, etc.) with a brief methodology description. According to some recent narrative reviews on this topic:

https://www.ncbi.nlm.nih.gov/books/NBK1186/

https://www.ncbi.nlm.nih.gov/pmc/articles/PMC4179092/

I am not sure what this article adds to current knowledge. The authors should have stated what was the main reason for writing such an article and what it adds to status because it is not clear to me.

Running a rapid review on this topic (with proper methodology: https://ebm.bmj.com/content/early/2023/04/19/bmjebm-2022-112079) would improve such work.

Response: Thank you very much for the constructive comment and suggestion. Compared to suggested papers (https://www.ncbi.nlm.nih.gov/books/NBK1186/; https://www.ncbi.nlm.nih.gov/pmc/articles/PMC4179092/) we provided updated information about pathogenesis, clinical and radiological features, diagnosis, and available treatment for CFC syndrome, suppling practice guidelines for the management of pediatric patient with CFC.

We stated at the end of Introduction section the aim of the present study in order to make it clearer to the readers: “The aim of the study is to give updated evidences on the pathogenesis, clinical and radiological features, diagnosis, and available treatment for CFC syndrome, suppling practice guidelines for the management of pediatric patient with CFC”.

The present study was designed to be a narrative review. We added a methodology section in the paper.

Minor Comments:

Abstract

I would add information that CFC syndrome is one of the rarest Rasopathies or I would add incidence of CFC or other Rasopathies

Response: Thank you for your comment. We added this information in the introduction section.

In the sentence “Phenotypic variability in CFC patients is strongly associated with different variants in these genes” I am not sure to what genes it refers to, probably this sentence should be used earlier in the article.

Response: Thank you for your comment. We corrected the sentence, highlighting how a phenotypic variability could be detected and related to the specific gene affected, even thought, there is no evidence of a genotype-phenotype correlation for specific pathogenic variants.

Please add what was the main reason for writing such an article and state what is new for the readers.

Response: Thank you for your comment. We answered to this point above. Specifically, we firstly mentioned in the review the new gene causally related to CFC, YWHAZ, describing how de novo missense pathogenic variants in this gene, led to RAS/MAPK pathway disruption and clinical manifestations (this gene was not reported in the latest CFC review, https://pubmed.ncbi.nlm.nih.gov/20301365/). Moreover, there are many additional features related to CFC that have emerged over the years. Therefore, we provided more updated information, compared to the review written in 2014 (https://pubmed.ncbi.nlm.nih.gov/25180280/).  

Neuroradiological findings

This is a well-described paragraph with a very large Figure 2. However, I am confused as to why is there such a large disproportion between the remaining paragraphs. Is that because the neurological changes are much more common in CFC syndrome?

Response: Thank you for your comment. This paragraph is longer than the others because includes both neurological and neuroradiological findings. Additionally, we focused more on neurological features because we found interesting phenotypic characteristics related to the specific gene affected. This sheds light on a potential future genotype-phenotype correlation with diagnostic-prognostic implications.

If the authors' aim was to summarize the clinical characteristics of CFC syndrome, I would suggest firstly balancing the volume of individual chapters and adding instead of figure 3 a real photo of patients CFC syndrome in different age (if it is possible for the authors)

Response: Thank you very much for your comment. We inserted photos of patients with CFC.

Current Consensus guidelines strategy include the following

Line 288 – “[4].” Should be without bold.

Response: Thank you very much for your suggestion. All reference numbers are in bold according to the Genes manuscript layout.

Treatment

This paragraph needs some improvements too. First of all, authors did not use up-to-date references according to NF1 and KRAS patients (some suggestions below). Secondly, I would focus more on studies in which the efficacy of MEK inhibitors was assessed in childrens or adults

Response: We are grateful for your comment. We updated the treatment section with the most recent evidences, as you suggested.

Reviewer 3 Report

Comments and Suggestions for Authors

The authors have conducted a comprehensive review of the current literature on Cardiofaciocutaneous (CFC) syndrome. The manuscript is well-structured and provides a broad and in-depth coverage of the topic. Here are a few suggestions for consideration:

  1. The authors have described the syndrome from a phenotypic perspective. Given that the journal, Genes, has a strong focus on genetics, it might be beneficial to include a table summarizing key genetic aspects of CFC syndrome. This could include variables such as gene name, type of inheritance, prevalence in CFC syndrome, phenotypic features, and corresponding OMIM entries.

  2. The authors have identified autosomal dominance as the primary pattern of inheritance in CFC syndrome. However, given the significant phenotypic severity of the disease, the likelihood of a disease-causing variant being passed on to the next generation seems low. Therefore, for the disease to remain stable within a certain population’s genetic pool, significant de novo mutations would need to occur in neonates. This phenomenon warrants further discussion.

  3. The infant depicted in Figure 3 does not appear to be representative of CFC syndrome and instead exhibits a more typical phenotype. It may be helpful to modify this figure to better illustrate the typical phenotype of CFC syndrome for readers.

Author Response

Reviewer 3

1) The authors have described the syndrome from a phenotypic perspective. Given that the journal, Genes, has a strong focus on genetics, it might be beneficial to include a table summarizing key genetic aspects of CFC syndrome. This could include variables such as gene name, type of inheritance, prevalence in CFC syndrome, phenotypic features, and corresponding OMIM entries.

Response: Thank you very much for your suggestion. We inserted the table. Of note, we only reported clinical features strictly related to each gene. Therefore, phenotypes not associated with a specific genotype were not mentioned, even if common.

2) The authors have identified autosomal dominance as the primary pattern of inheritance in CFC syndrome. However, given the significant phenotypic severity of the disease, the likelihood of a disease-causing variant being passed on to the next generation seems low. Therefore, for the disease to remain stable within a certain population’s genetic pool, significant de novo mutations would need to occur in neonates. This phenomenon warrants further discussion.

Response: Thank you for your comment. We inserted a discussion of this point in the “Diagnosis” section, to highlight the rare possibility of a vertical transmission of CFC pathogenic variants, suggesting that these ones are compatible with human reproduction. However, most individuals present the disorder as the result of a de novo pathogenic variants within the Ras/MAPK pathway.

3) The infant depicted in Figure 3 does not appear to be representative of CFC syndrome and instead exhibits a more typical phenotype. It may be helpful to modify this figure to better illustrate the typical phenotype of CFC syndrome for readers.

Response: Thank you very much for your comment. We inserted representative photos of CFC patients.

Round 2

Reviewer 2 Report

Comments and Suggestions for Authors

I have no other comments. I am satisfied with the changes introduced by the authors. In my opinion, the manuscript has been significantly improved

Author Response

I have no other comments. I am satisfied with the changes introduced by the authors. In my opinion, the manuscript has been significantly improved

Response: Thank you very much. We are grateful for your comment.

Reviewer 3 Report

Comments and Suggestions for Authors

I have re-evaluated the manuscript and overall, the authors have revised it well. However, one remaining concern is the images presented in Figure 3. These images may infringe on the patients’ privacy as they are not de-identified, and I could not find a specific statement indicating that the authors obtained informed consent from the patients.

Author Response

I have re-evaluated the manuscript and overall, the authors have revised it well. However, one remaining concern is the images presented in Figure 3. These images may infringe on the patients’ privacy as they are not de-identified, and I could not find a specific statement indicating that the authors obtained informed consent from the patients.

Response: Thank you very much for your comment. We inserted in the methodology section that we obtained informed consent from the family patient. Additionally, we inserted it as supplementary file.